# Health professionals' readiness to implement electronic medical recording system and associated factors in public general hospitals of Sidama region, Ethiopia

**Kibruyisfaw Weldeab Abore**[1]*, **Alemu Tamiso Debiso**[2], **Betelhem Eshetu Birhanu**[2], **Bezahegn Zerihun Bua**[3], **Keneni Gutema Negeri**[2]

1 Department of Pediatrics, Yirgalem Hospital Medical College, Yirgalem, Sidama, Ethiopia, 2 School of Public health, College of Medicine and Health Science, Hawassa University, Hawassa, Ethiopia, 3 Health system strengthening project, College of health science, Hawassa University, Hawassa, Ethiopia

* Kibruyisfaww@gmail.com

**Data Availability Statement:** All relevant data are within the paper and its Supporting Information files.

## Abstract

### Background

Electronic medical recording system is one of the information technologies that has a proven benefit to improve the quality of health service. Readiness assessment is one of the recommended steps to be taken prior to implementing electronic medical recording system to reduce the probability of failure.

### Objective

To determine the level of health professional readiness to implement Electronic medical recording system and associated factors in public general hospitals of Sidama region, 2022.

### Methodology

A cross-sectional study design complemented with qualitative study was employed at three public general hospitals in Sidama region on a sample of 306 participants. A pretested self-administered questionnaire was used to collect quantitative data and in-depth interview was used for the qualitative study. Bivariate and multivariate Binary logistics regression was performed to determine predictors of readiness at α = 0.05, using an odds ratio and 95% confidence interval. Thematic analysis was done for qualitative data collected through in-depth interview.

### Result

The overall readiness for health professionals was 36.5%. Of the study participants, 201 (73.4%) were computer literate, 176(64.23%) had good knowledge, and 204 (74.45%) had favorable attitude towards EMR. Only 31 participants had previous training (11.3%), while 64 (23%) had previous experience. EMR knowledge (AOR = 3.332; 95%CI: (1.662, 6.682)) and attitude towards electronic medical recording (AOR = 2.432; 95%CI: (1.146, 5.159))

**Funding:** The funding for this study was obtained from the Doris Duke Charitable foundation data usage partnership project research grant (grant number CHIMEA046). K.W.A received the award. The funder had no role in study design, data collection and analysis, decision to publish, or preparation of the manuscript.

**Competing interests:** The authors have declared that no competing interests exist.

were statistically significant predictors of readiness to implement electronic medical recording. Qualitative analysis has revealed lack of training, ease of use concerns, information security concerns, and perceived inadequacy of infrastructures including internet connectivity and electricity as common barriers for health professional readiness to implement EMR.

## Conclusion

Health professionals' readiness in this study was low. Capacity building efforts to increase the awareness and skills of health professionals should be done before implementing the system.

## Introduction

The demand for technologies that can accommodate for the large volume of information generated by the health care system has increased in the 21$^{st}$ century. This need had motivated countries to adopt an innovative way of handling medical records known as electronic medical recording (EMR). EMR system is a digital health technology utilized in the health sector to collect, generate and present health related data by the health professionals along with exchanging information with authorized personnel within the health care setting [1].

EMR has proven benefits to improve the quality of service by improving efficiency and productivity through timely decision-making, saving recurring costs, reducing medical errors, increasing patients' safety, ensuring data confidentiality, and sharing medical information between authorized personnel [2–5]. These benefits are more pronounced in developing and low income areas like sub-Saharan countries which are constantly ravaged by pandemics and epidemics [6,7].

Globally, less than half of the world countries has adopted a national EMR system according to a 2016 WHO report, although there have been improvements over the past decade [1,8]. The report had also shown disparities in the adoption of the system among countries. Globally, Israel, Canada, Denmark, and Australia had notable achievements in the implementation of electronic health records [9]. However, the adoption of EMR system in developing countries was low. This low level of implementation is attributed to the high level of both budget and human resources required by the system which includes large financial investment, better infrastructures including electricity and internet connectivity, and skilled manpower [10,11].

For the same obvious reasons and exacerbated by poor infrastructures, African continent and the sub-Saharan countries in particular are lagging behind the world in implementing EMR, in spite of the high disease burden and health demand [6,10,12].

Although Health information system has been used for long period to generate aggregate data to be used at different administrative levels, the implementation of EMR in Ethiopia is still young. EMR with a name of smartcare, which was later called Tenacare, was first piloted in Ethiopia in 2009 by the Ministry of Health (MOH) with a support from Tulane University technical assistance project in Ethiopia (TUTAPE) [13,14]. Currently, the role of digital technologies is given great emphasis after the MOH acknowledged benefits of digitization and the ministry set out to change the culture of information generation and utilization for evidence based decision making at all levels[14]. The quality and availability of health information would significantly impact the quality of health care provision. Individual level data including but not limited to demographic, clinical, laboratory investigation, imaging, and medication history and billing were targeted [14,15].

Readiness assesses the level of preparedness and how welcoming a given institution and its professionals will be to the changes brought by adopting a new technology [16,17]. The need for this pre-implementation assessment emanates mainly from the resource intensive nature of the process and its dependence on human and organizational factors for its success; equally to the technical aspect. The probability of failure of EMR system could be minimized if an appropriate pre-implementation assessment of readiness is done and the concerns and gaps of practitioners are addressed properly [18]. A readiness assessment is paramount to save the unnecessary expenditure of energy, time, and money.

There are few published studies done in Ethiopia to assess health professional readiness and there was no study conducted in the region to best of the investigators knowledge [13,15]. Thus, this study aims to determine the level of health professional readiness to implement Electronic medical recording system and associated factors in public general hospitals of Sidama region.

## Methods and materials

### Study setting and study design

Sidama region is one of the 11 regions in Ethiopia. Based on up to date information there are four general hospitals in the region namely Adare, Yirgalem, Bona, and Leku which are found 275 km, 325km, 392km, and 306 km away from the capital Addis Ababa respectively. Adare hospital was excluded from the study since EMR system is already implemented. Institution based cross-sectional study complemented with qualitative study was conducted from April 15 –May 10, 2022.

### Study population and inclusion criteria

Health professionals working at the three selected general hospitals of Sidama region were the study population. Professionals who have worked for six months or more at their respective hospitals were included to ensure adequate exposure has occurred to medical record keeping.

### Sample size and sampling procedure

Sample size was determined using Epi info version 7 using the following assumptions: 80% power, proportion of readiness among those with favorable attitude 0.75 [15], AOR = 1.63 [15], 95% confidence interval. Final sample size of 306 was achieved after accounting for finite population correction and 10% non-response. After allocating proportional sample size to each hospital, Stratified random sampling was utilized to reach the final sample using a sampling frame containing the list of professionals in each professional category of each hospital. For the qualitative study 4 key informants from each hospital were purposively selected and data was collected until saturation of information was achieved.

### Data collection technique and quality control

A structured self-administered questionnaire adapted after reviewing literatures and translating in to Amharic language was utilized for data collection. The tool was pretested on 5% of the final sample size (16 professionals) at Tula primary hospital, which has a near similar setup to the study area. The reliability of the tool that was used to assess readiness was tested and Cronbach alpha for core readiness was 0.763 and for engagement readiness it was 0.712. The tool was declared reliable as the result was > 0.7. Data collection was done by three trained nurses and supervision was done by a Masters student with previous experience of research. In-depth interview with medical directors, nurse matrons, quality unit heads and health

management information system focal person through Amharic language was conducted to collect qualitative data. Among the interviewees, 8 of them were male while 4 of them were females. Interviewees were consisted of 6 medical doctors, 3 nurses, and 3 health information technology professionals. The supervision of data collection and quality control was done by the supervisor and the primary investigator at each hospital.

## Operational definition

Computer literacy was measured using a set of self-assessment questions regarding the responders' ability to perform routine tasks on a computer [19]. Professionals who scored $\geq$ 50% on literacy questions were classified as computer literate.

Knowledge is measured as a latent variable of a set of five questions which assessed whether the individual has the basic knowledge about EMR. professionals that scored 50% or more for the knowledge questions were said to have good knowledge [15].

Attitude was measured as a latent variable of a set of six questions that assesses the individual perception of EMR measured on a five point Likert scale. A score of median or above was used to classify as having a favorable attitude [15].

Core readiness was measured as a latent variable of a set of four questions measured on five point Likert scale based on Li Et al. [17] that addressed satisfaction with the current paper based system and the desire for change. A professional who scored above or equal to the median were labeled to have core readiness [13,15]. Meanwhile, Engagement readiness was measured as a latent variable of a set of nine questions measured on five point Likert scale based on Li Et al. [17] that addressed the willingness to use EMR and the professionals perceived benefits and harms of EMR. A professional who scored above or equal to the median were labeled to have engagement readiness [13,15]. Health professionals who have both core readiness and engagement readiness were labeled to have an overall readiness [17].

## Data processing and analysis

Data entry, coding, and verification were done using epi-data 3.1. After exporting the data analysis was done using SPSS version 20. Categorical data were summarized using frequency and percentages. Simple binary logistic regression was performed to assess predictors of readiness and those variables with a p-value of <0.25 were considered as candidate for multivariable Logistic regression to determine predictors of readiness, using $\alpha$ = 0.05 as the significance level. Association was measured using Odds ratio with the corresponding 95% confidence interval. Qualitative data from audio recording was transcribed into Amharic and translated to English. After importing the text file, coding was done supported by Atlas.ti version 7.5.7 software. The codes were further categorized into themes and subthemes after which Thematic analysis was done.

## Ethical statement

Ethical approval (approval number IRB/059/14) was obtained from Hawassa University, College of medicine and health science institution review board. A formal letter from the university addressed to the participating hospitals was taken and submitted. Informed written consent was obtained from the participants after thoroughly discussing the idea behind the study, and study participant rights. Participants were also assured that the confidentiality of the information they provided would be maintained.

## Result and discussion

Of the total 306 participants 274 returned the questionnaire with a response rate of 89.5%.

## Sociodemographic characteristics

In this study, more than half of the participants (53.3%) were within the age category 25–29 and 177 (64.6%) of the participants were male professionals. About 121 (44.2%) respondents were nurses, 40(14.6%) were doctors, 27(9.9%) were midwives, 27(9.9%) were pharmacists and 26(9.5%) were laboratory technicians. Of the respondents 192 (70.1%) have a bachelor degree while 21(7.7%) health professionals had a Masters degree and above. Moreover, 176 (64.2%) of respondents have served at the hospital where they are currently working (Table 1).

## Organizational and technical factors

It was found that 126 (46%) of respondents have a personal computer at home. It was also found that 201 (73.4%) of the participants were computer literate. With regard to computer use, 158 (70.2%) of the participating professionals used computers for both work and entertainment purposes. among the study participants, only 31 (11.3%) had previous EMR training while only 63 (23%) had previous experience using EMR system. Of the 274 health professionals, 110 (40.1%) of them said they have computer access at workplace while 161 (58.8%) of the respondents also said they don't have internet access at workplace. In addition, Only 92

**Table 1. Sociodemographic characteristics of health professionals working in general hospitals, Sidama region 2022.**

| Variables | | Frequency (N = 274) | Percentage |
|---|---|---|---|
| Age group | | | |
| | 20–24 | 37 | 13.5 |
| | 25–29 | 146 | 53.3 |
| | 30–34 | 75 | 27.4 |
| | ≥35 | 16 | 5.8 |
| Sex | | | |
| | Male | 177 | 64.6 |
| | Female | 97 | 35.4 |
| Profession | | | |
| | Nurses | 121 | 44.2 |
| | Midwife | 27 | 9.9 |
| | Pharmacists | 27 | 9.9 |
| | Laboratory | 26 | 9.5 |
| | Doctors | 40 | 14.6 |
| | Health officer | 10 | 3.6 |
| | Health information technologists | 8 | 2.9 |
| | Others | 15 | 5.5 |
| Education status | | | |
| | Diploma | 61 | 22.3 |
| | Degree | 192 | 70.1 |
| | Second degree and above | 21 | 7.7 |
| Duration of service at current hospital | | | |
| | 6–12 months | 43 | 15.7 |
| | 13–18 months | 26 | 9.5 |
| | 19–24 months | 29 | 10.6 |
| | More than 24 months | 176 | 64.2 |

Others; anesthetists, radiographers, integrated emergency surgery officers, environmental health.

(33.6%) health professionals believe that their hospital has adequate infrastructure. Moreover, only 91 (33.2%) health professionals think there would be strong managerial support if EMR is implemented at their hospitals (Table 2).

### Health professionals knowledge and attitude towards EMR system

Regarding knowledge about EMR, 176 (64.2%) of respondents had good knowledge. Meanwhile, 204 (74.5%) respondents had favorable attitude toward the EMR system (Fig 1).

### Readiness to implement EMR system

The core readiness to implement EMR in this study was 55.8% and the engagement readiness was 54%. Of the study participants, only 100 (36.5%) had overall readiness and were ready to use EMR (Fig 2). This is significantly lower than what studies conducted in other parts of Ethiopia 62.3% [15], Ghana 54.9% [20], and Myanmar 54.2% [19] reported. However, this difference could also be due to method used to classify the readiness of professionals, differences in sample size or differences in sociodemographic characteristics.

### Factors associated with readiness to implement EMR system

In this study, duration of employment, knowledge about EMR, and attitude towards EMR were found to be statistically significant predictors of health professional readiness after adjusting for other variables. It was found that those health professionals who have worked for 13 to 18 months at the hospital where they are currently working had 3.85 times higher odds of being ready than those who have worked for more than 24 months (AOR = 3.848, 95% CI; (1.428,10.371)). This result differs from a study done in Ghana which showed old employees to be more likely to be ready than new employees [20]. The finding could be explained by the fact that early level employees are young professionals who can easily utilize technologies while also being sufficiently exposed to medical recording system [21].

Among health professionals, those who had good EMR knowledge had 3.33 times higher odds of being ready than those with poor knowledge (AOR = 3.332, 95% CI; (1.662, 6.682)). This finding is supported by studies done in other parts of Ethiopia [13,15], Ghana [20], and Myanmar [19]. This can be explained by the fact that a professional having good knowledge about EMR could have higher chance of understanding about the potential benefits that the system would bring to the professionals, the patients, and the overall service. This finding is also supported by qualitative study results.

A 35 year old participant said "I didn't have any training or previous experience working with EMR. While I was attending a relative, I have seen it being practiced in private healthcare settings and I was able to see its benefits. It had made me eager to know more about the system"

It was also noted that those professionals with favorable attitude had 2.43 times higher odds of being ready than professionals with unfavorable attitude (AOR = 2.432, 95% CI; (1.146, 5.159)) (Table 3). This could be explained by the fact that professionals could likely be willing to use the system if they have a favorable and positive image with good interest towards the system. Previous studies had also shown that health professional's attitude affects not only their readiness but also the actual utilization of the system [22]. The finding is also supported by qualitative study results.

A 30 year old participant said "I do not think there would be a problem for me to use the system effectively. If I am instructed on a few things and if I am provided with the software, I think I would build upon what I know and be better able to use it."

**Table 2. Technical and organizational factors for health professional readiness in Sidama region.**

| Variables | | Frequency | Percentage |
|---|---|---|---|
| Have Personal computer at home | | | |
| | Yes | 126 | 46 |
| | No | 148 | 54 |
| Ever used computer | | | |
| | Yes | 225 | 82.1 |
| | No | 49 | 17.9 |
| Duration of computer use | | | |
| | Less than 6months | 32 | 14.2 |
| | 6–12 months | 37 | 16.4 |
| | 13–24 months | 24 | 10.7 |
| | More than 24 months | 132 | 58.7 |
| Purpose of computer use | | | |
| | Work purpose only | 54 | 24 |
| | Entertainment purpose only | 13 | 5.8 |
| | Work and entertainment | 158 | 70.2 |
| Computer Literacy | | | |
| | Literate | 201 | 73.4 |
| | Illiterate | 73 | 26.6 |
| Previous EMR training | | | |
| | Yes | 31 | 11.3 |
| | No | 243 | 88.7 |
| Previous EMR experience | | | |
| | Yes | 63 | 23 |
| | No | 211 | 77 |
| Workplace computer access | | | |
| | Yes | 110 | 40.1 |
| | No | 164 | 59.9 |
| Workplace internet access | | | |
| | Yes | 161 | 58.8 |
| | No | 113 | 41.2 |
| Adequate infrastructures | | | |
| | Yes | 92 | 33.6 |
| | No | 182 | 66.4 |
| Adequate management support | | | |
| | Yes | 91 | 33.2 |
| | No | 183 | 66.8 |

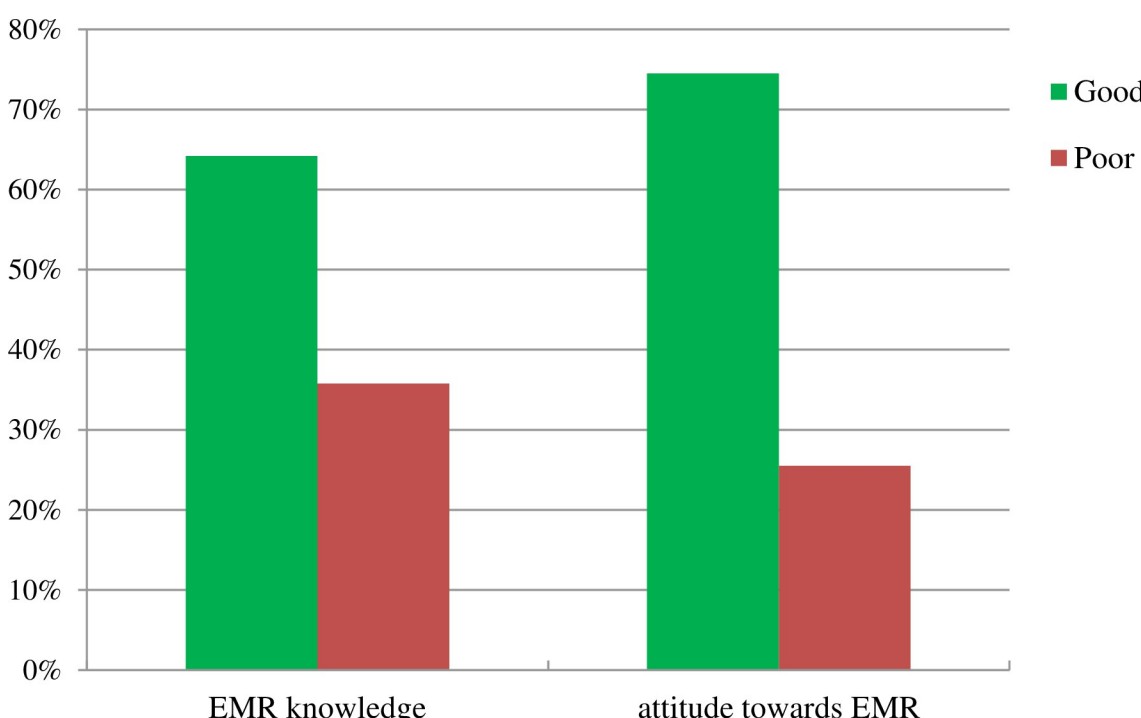

**Fig 1. EMR knowledge and attitude towards EMR among health professionals working in public general hospitals, Sidama, Ethiopia, 2022.**

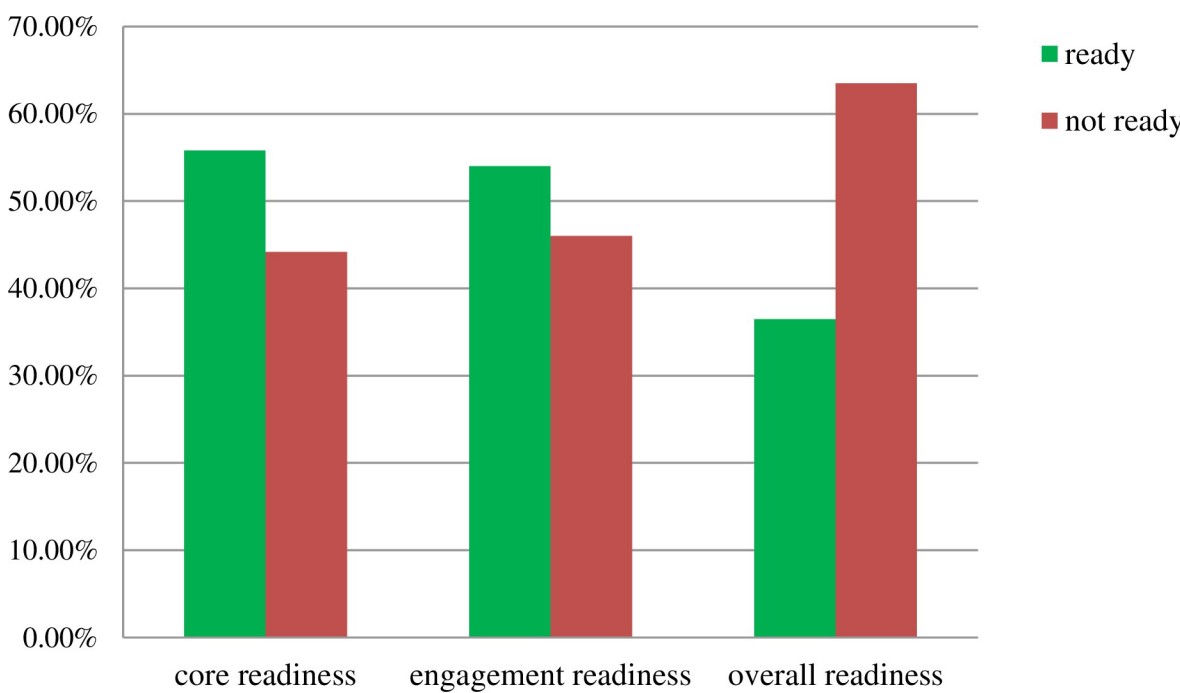

**Fig 2. Health professional readiness to implement EMR in public general hospitals of Sidama region, Ethiopia, 2022.**

**Table 3. Bivariable and multivariable analysis of factors associated with health professional readiness to implement EMR in public general hospitals of Sidama region, 2022.**

| Variables | | Not ready (174, 63.5%) | Ready (100, 36.5%) | COR (95% C.I) | AOR(95% CI) | P-value |
|---|---|---|---|---|---|---|
| Duration of employment | | | | | | |
| | 6–12 months | 32(74.4%) | 11(25.6%) | 0.573 (0.27,1.21) | 0.797 (0.321, 1.977) | 0.624 |
| | 13–18 months | 11(42.3%) | 15(57.7%) | 2.273 (0.985,5.242) | 3.848 (1.428, 10.371) | 0.008* |
| | 19–24 months | 21(72.4%) | 8(27.6%) | 0.635 (0.266,1.515) | 0.931 (0.339, 2.557) | 0.89 |
| | > 24 months | 110(62.5%) | 66(37.5%) | 1.00 | 1.00 | |
| EMR Knowledge | | | | | | |
| | good | 94(53.4%) | 82(46.6%) | 3.877 (2.147,7.0) | 3.332 (1.662,6.682) | 0.001* |
| | poor | 80(81.6%) | 18(18.4%) | 1.00 | 1.00 | |
| Attitude towards EMR | | | | | | |
| | Favorable | 116(56.9%) | 88(43.1%) | 3.667 (1.857,7.241) | 2.432 (1.146,5.159) | 0.021* |
| | Unfavorable | 58(82.9%) | 12(17.1%) | 1.00 | 1.00 | |

Variables accounted for: Age category, profession category, previous EMR experience, ownership of personal computer, workplace computer access, perceived adequacy of infrastructure, and perceived management support.

*P<0.05.

## Perceived barriers for readiness to implement EMR

Qualitative analysis from the in-depth interview conducted on participants with mean age of 28 to explore barriers for readiness to implement EMR had shown that majority of participants had concerns pertaining to lack of training, concerns related EMR system ease of use, patients information security concerns, and perceived inadequacy of infrastructures including internet connectivity and electricity as common barriers for health professional readiness to implement EMR.

## Training related

Among the participating hospitals, Leku hospital has already done facility assessment and installed servers needed, although professionals hadn't been trained. Most of the participants predominantly raised their concern about lack of training regarding EMR system and computer related skills. They also discussed the need for continuous on job orientations, monitoring, and follow up. Furthermore, one participant discussed the need to avail manuals by the professional's side. A 32 year old participant explained the need for a skill laboratory for professionals to improve their skills.

He stated "*If professionals who are not familiar with computers want to use the system, it would be difficult for them. Once the system is implemented, we need to have skill labs where professionals could develop their computer skills to better understand the system*".

**System related.** Four of the study participants expressed concerns regarding the EMR system itself. Issues related to security including hacking, the dependence of the system on stable connection and electricity, and issues of maintenance were some of the concerns raised. Furthermore, one participant explained the need for parallel documentation to better secure information. A 32 year old male interviewee explained

"*The software can be attacked by different things. One thing is it can be corrupted or hacked and it can take away all patients data. I think it would be better if there is a printed hard copy of everyday records*".

**Facility related.**   Respondent also raised their concerns regarding the adequacy of infrastructures in their institution. Issues of electricity, internet connectivity, and availability of computers at work stations were raised predominantly. It was also noted that those concerns were shared by all participants of the interview. Participants suggested this issue should be properly dealt with prior to implementation. A 28 year old female participant denoted

"*The stability of the internet connection around here is concerning. If the system is implemented without addressing this, it can frustrate the professional and might eventually lead to prefer the paper based record system*"

## Strength and limitation

This study is the first study done in the region to assess the readiness of health professionals to implement EMR. The study utilized mixed method design which enabled to better explore the level of readiness and factors related with its implementation. However, the study was not without limitations. First, data was collected over a period of 3 days per institution. This might have led to information sharing among participants regarding the questions raised in the questionnaire and affected the assessment of knowledge level. Secondly, although the study utilized composite questions to assess computer literacy, the assessment of computer literacy was subjective. Thus, the high level of computer literacy reported in this study might not reflect the truth and needs objective assessment.

## Conclusion

Overall, the readiness of health professionals to implement EMR in this study was low. Duration of employment, knowledge about EMR, and attitude towards EMR were found to be statistically significant predictors of readiness.

## Recommendation

Capacity building and awareness creation efforts including training should be provided to health professionals prior to implementation to increase the level of knowledge about EMR among health professionals. Presumably, this could also change the attitude of health professionals as it would increase the skills of professionals and it would make them feel competent and willing to use the system. Furthermore, further study is recommended to assess the factors that affect the knowledge and attitude of health professional towards EMR system.

## Supporting information

**S1 Data. Minimal data.**
(XLSX)

## Acknowledgments

We would also like to thank Hawassa University for providing us the ethical clearance for this study. We would also like to show our heartfelt gratitude for the participating health institutions involved in this study.

## Author Contributions

**Conceptualization:** Kibruyisfaw Weldeab Abore.

**Data curation:** Kibruyisfaw Weldeab Abore, Bezahegn Zerihun Bua.

**Formal analysis:** Kibruyisfaw Weldeab Abore, Alemu Tamiso Debiso, Betelhem Eshetu Birhanu, Keneni Gutema Negeri.

**Funding acquisition:** Kibruyisfaw Weldeab Abore.

**Investigation:** Kibruyisfaw Weldeab Abore.

**Methodology:** Kibruyisfaw Weldeab Abore, Alemu Tamiso Debiso, Betelhem Eshetu Birhanu, Bezahegn Zerihun Bua, Keneni Gutema Negeri.

**Project administration:** Kibruyisfaw Weldeab Abore.

**Resources:** Alemu Tamiso Debiso.

**Software:** Kibruyisfaw Weldeab Abore, Bezahegn Zerihun Bua.

**Supervision:** Alemu Tamiso Debiso, Betelhem Eshetu Birhanu, Keneni Gutema Negeri.

**Validation:** Betelhem Eshetu Birhanu, Bezahegn Zerihun Bua, Keneni Gutema Negeri.

**Visualization:** Kibruyisfaw Weldeab Abore.

**Writing – original draft:** Kibruyisfaw Weldeab Abore.

**Writing – review & editing:** Alemu Tamiso Debiso, Betelhem Eshetu Birhanu, Bezahegn Zerihun Bua, Keneni Gutema Negeri.

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
