## [Decision Letter · Decision Letter 0]

1 Aug 2022

PONE-D-22-16693Health professionals’ readiness to implement electronic medical recording system and associated factors in public general hospitals of Sidama region, EthiopiaPLOS ONE

Dear Dr. Abore,

Thank you for submitting your manuscript to PLOS ONE. After careful consideration, we feel that it has merit but does not fully meet PLOS ONE’s publication criteria as it currently stands. Therefore, we invite you to submit a revised version of the manuscript that addresses the points raised during the review process.

We look forward to receiving your revised manuscript.

Kind regards,

Humayun Kabir, MSc in Epidemiology

Academic Editor

PLOS ONE

Journal Requirements:

Reviewers' comments:

Reviewer's Responses to Questions

**Comments to the Author**

1. Is the manuscript technically sound, and do the data support the conclusions?

Reviewer #1: Partly

Reviewer #2: Partly

2. Has the statistical analysis been performed appropriately and rigorously? 

Reviewer #1: Yes

Reviewer #2: No

3. Have the authors made all data underlying the findings in their manuscript fully available?

Reviewer #1: No

Reviewer #2: No

4. Is the manuscript presented in an intelligible fashion and written in standard English?

Reviewer #1: Yes

Reviewer #2: Yes

5. Review Comments to the Author

**Reviewer #1: **Abstract: The result doesn’t reflect anything from qualitative section. No need to mention the name of Software used in data analysis.

Introduction: Kindly reflect how EMR system could contribute to better data management and what kind of health data are mostly aimed to handle by EMR system in Ethiopia.

Methods: Provide justification why p value less than 0.25 was considered as candidate for multivariate Logistic regression to determine predictors of readiness. On what basis were the professionals who scored ≥ 50% on literacy questions classified as computer literate? Mention the process of qualitative data analysis. Focus on preconceptions and meta-positions for qualitative data analysis process.

Results and discussions: Although the result fairly comply with the quantitative analysis, it seriously lacks the qualitative findings. There are only two sentences from the participants which does not show the strength of the qualitative data analysis. What theme was developed from the thematic analysis? The discussion should be more detailed and should focus on validity and reliability for quantitative findings and transferability for qualitative findings.

Recommendations and conclusions could be separate sections.

**Reviewer #2:** This study investigated the health professional readiness to implement electronic medical recording system and associated factors in public general hospitals in a region in Ethiopia. The idea was good, and the findings of this research could help the policymakers in future decision-making.

However, after carefully reading the whole manuscript, I feel that this manuscript needs a major revision to achieve the merit of publication in a high-quality journal.

Overall comments:

1. There are a few grammatical and punctuational errors in the manuscript. This should be addressed in the revised version.

Abstract:

2. “Electronic medical recording system is one of the information technologies that have a proven benefit to improve the quality of health service.” It will be “has a proven benefit”.

3. “A hospital based cross-sectional study design”; remove the term “hospital based”, as in the subsequent texts, it is mentioned that the data was collected from the hospital.

Introduction:

4. “Globally, less than half of the world countries has adopted a national EMR system according to a 2016 WHO report, although there have been improvements over the past decade [1, 8]. The report had also shown disparities in the adoption of the system among countries. A 2013 report indicated that more than 90% of primary care physicians in New Zealand, Netherlands, United Kingdom, Australia, Sweden, and Norway used EMR system [9].” These are old statistics. Please report the updated statistics.

5. MOH: use the full form of its first appearance.

6. “There are few published studies done in Ethiopia to assess health professional readiness and there was no study conducted in the region to best of the investigators knowledge.” Use citations.

7. Mention the objective of the study at the end of the introduction.

Methods and materials:

8. “Based on up to date information there are four general hospitals in the region namely Adare, Yirgalem, Bona, and Leku which are found 275 km, 325km, 392km, and 306 km away from the capital Addis Ababa respectively.” Use citations.

9. The authors mentioned G.C at the end of the date. What does it mean?

10. What are the types of health professionals included in the study? It should be mentioned in the study population section.

11. “12 key informants” from how many hospitals? From any single or from all hospitals?

12. In the logistic regression, it will multivariable analysis. Not multivariate analysis, as there is dichotomous dependent variable.

13. Operational definitions should be placed before data processing and analysis section.

Results:

14. “Of the respondents 192 (70.1%) have a first degree while 21(7.7%) health professionals had a second degree and above.” What are first degree and second degree? Not clear. Explain in the methods section or in the result interpretation.

15. In table 1, what is HIT profession?

16. In table 1, What does mean by degree and second degree professions?

17. “among the study participants, only 31 (11.3%) have previous training while only 63 (23%) had previous experience.” Write in the past tense.

18. In table 2, the variables ‘previous training’ and ‘previous experience’ are not clear. What training and what experience?

19. In table 3, p value placement of ‘duration of employment’ is not correct. The authors put value only for unadjusted analysis. Why not for adjusted analysis? Put p value for both the analyses, or indicate it using asterisk.

20. It is not clear how the authors adjusted the readiness of EMR use in the regression model. It should be adjusted by including some important demographic variables also, i.e. age, sex, educational status, profession etc.

References:

21. For most of the citations, there is no doi or url in the references.

22. There is no data as supplementary file.

6. PLOS authors have the option to publish the peer review history of their article (what does this mean?). If published, this will include your full peer review and any attached files.

Reviewer #1: No

Reviewer #2: **Yes: **Saifur Rahman Chowdhury

---

## [Author Response · Author response to Decision Letter 0]

29 Aug 2022

We would like to take a moment to thank the Editor and the peer reviewers for the constructive evaluation of our paper. We have corrected the manuscript as per the comments provided and we hope the manuscript meets the requirement for publication.

Sincerely,

Kibruyisfaw Weldeab Abore

Response to Academic editor

We have edited authors contributorship as per the journal requirement. We have also edited the figure files using PACE and revised the file names to suit the journal. 

We have amended the information on both sections

3. Upon re-submitting your revised manuscript, please upload your study’s minimal underlying data set as either Supporting Information files or to a stable, public repository and include the relevant URLs, DOIs, or accession numbers within your revised cover letter.

Thank you for the comments. We have availed the minimal data set as a supporting information.

 

Reviewer 1

Abstract: The result doesn’t reflect anything from qualitative section. No need to mention the name of Software used in data analysis

We amended abstract section to include results from the qualitative analysis and we have removed the name of the software used in data analysis from the abstract.

Introduction: Kindly reflect how EMR system could contribute to better data management and what kind of health data are mostly aimed to handle by EMR system in Ethiopia

We have amended the manuscript to explain the kind of health data intended to be handled by EMR and the role of EMR for better data management.

Methods: Provide justification why p value less than 0.25 was considered as candidate for multivariate Logistic regression to determine predictors of readiness. On what basis were the professionals who scored ≥ 50% on literacy questions classified as computer literate? Mention the process of qualitative data analysis. Focus on preconceptions and meta-positions for qualitative data analysis process

Although different scholars use p-value < 0.2 for including predictors in to the multivariable logistic regression based on Hosmer and lemshow applied logistic regression recommendation, we used p-value< 0.25 to include as many predictor variables as possible in the multivariable model so as to ensure adjustment of confounders. Regarding computer literacy, we classified those participants with the average computer literacy score or above as literate. The questions were composed of questions which assessed basic computer skills of individuals. Pertaining to the process of qualitative analysis we have included details in the method and material section to meet reviewers comment.

Results and discussions: Although the result fairly complies with the quantitative analysis, it seriously lacks the qualitative findings. There are only two sentences from the participants which do not show the strength of the qualitative data analysis. What theme was developed from the thematic analysis? The discussion should be more detailed and should focus on validity and reliability for quantitative findings and transferability for qualitative findings

We have added a subsection labeled perceived barriers for readiness to implement EMR to address themes and subthemes that emerged during thematic analysis.

Recommendations and conclusions: could be separate sections.

We have amended this section in to two separate sections.

Reviewer 2.

Abstract:

1. There are a few grammatical and punctuation errors in the manuscript. This should be addressed in the revised version.

Thank you for the comments. We have amended the manuscript based on the comments.

2. “Electronic medical recording system is one of the information technologies that have a proven benefit to improve the quality of health service.” It will be “has a proven benefit”.

Thank you for the comment. We have addressed the comment 

3.“A hospital based cross-sectional study design”; remove the term “hospital based”, as in the subsequent texts, it is mentioned that the data was collected from the hospital.

Thank you for the comment. We have addressed the comment 

Introduction:

4.“Globally, less than half of the world countries has adopted a national EMR system according to a 2016 WHO report, although there have been improvements over the past decade [1, 8]. The report had also shown disparities in the adoption of the system among countries. A 2013 report indicated that more than 90% of primary care physicians in New Zealand, Netherlands, United Kingdom, Australia, Sweden, and Norway used EMR system [9].” These are old statistics.Please report the updated statistics.

Thank you for the comments. We have amended the section to include more updated information 

5.MOH: use the full form of its first appearance.

Thank you for the comment. We have addressed the comment 

6.“There are few published studies done in Ethiopia to assess health professional readiness and there was no study conducted in the region to best of the investigators knowledge.” Use citations.

Thank you for the comment. We have addressed the comment in the introduction section. 

7.Mention the objective of the study at the end of the introduction.

Thank you for the comment. We have addressed the comment at the end of the introduction section. 

Methods and materials:

8. “Based on up to date information there are four general hospitals in the region namely Adare, Yirgalem, Bona, and Leku which are found 275 km, 325km, 392km, and 306 km away from the capital Addis Ababa respectively.” Use citations.

Thank you for the comment.

9. The authors mentioned G.C at the end of the date. What does it mean?

Thank you for the comment. It means Gregorian calendar while in Ethiopia we use Ethiopian calendar ( E.C). we have removed all references of G.C from the manuscript.

10. What are the types of health professionals included in the study? It should be mentioned in the study population section.

Thank you for the comment. All health professionals working in the selected health facilities were considered the study population. Meanwhile, the details of the professional category were addressed in the sociodemographic part of the result. 

11. “12 key informants” from how many hospitals? From any single or from all hospitals?

A total of 12 key informants were selected from all hospitals. We have amended the manuscript to address the concern.

12. In the logistic regression, it will multivariable analysis. Not multivariate analysis, as there is dichotomous dependent variable.

Thank you for the valuable comment. We have addressed the comment at the method and material section and result section. 

13. Operational definitions should be placed before data processing and analysis section.

Thank you for the comment. We have placed operational definitions before data processing and analysis section.

Results

14. “Of the respondents 192 (70.1%) have a first degree while 21(7.7%) health professionals had a second degree and above.” What are first degree and second degree? Not clear. Explain in the methods section or in the result interpretation.

Thank you for the comment. We have addressed the issue in the result section. We have changed it to bachelor degree and Masters degree and above.

15. In table 1, what is HIT profession?

It means Health information technology professionals. We have amended it in the result section

16. In table 1, What does mean by degree and second degree professions?

Thank you for the comment. We have addressed the issue in the result section. We have changed it to bachelor degree and Masters degree and above.

17. “among the study participants, only 31 (11.3%) have previous training while only 63 (23%) had previous experience.” Write in the past tense.

Thank you for the comment. We have addressed the issue in the result section.

18. In table 2, the variables ‘previous training’ and ‘previous experience’ are not clear. What training and what experience?

Thank you for the comment. We have addressed the issue in the result section to previous EMR experience and previous EMR training.

19. In table 3, p value placement of ‘duration of employment’ is not correct. The authors put value only for unadjusted analysis. Why not for adjusted analysis? Put p value for both the analyses, or indicate it using asterisk.

Thank you for the comment. We have addressed the issue in the result section and we have indicated significant p-values using asterisk.

20. It is not clear how the authors adjusted the readiness of EMR use in the regression model. It should be adjusted by including some important demographic variables also, i.e. age, sex, educational status, profession etc.

Thank you for the comment. Although in the result section we have only displayed those with significant p-value on adjusted analysis, we did the multivariable logistic regression including variables from sociodemographic characteristics and organization and technical factors to adjust for confounders. If needed we can provide the detailed crude analysis and adjusted analysis results.

References

21. For most of the citations, there is no doi or url in the references.

Thank you for the comments. We have updated the references to include doi and url

22. There is no data as supplementary file.

Thank you for the comments. We have included minimal data as a supplementary file

---

## [Decision Letter · Decision Letter 1]

29 Sep 2022

PONE-D-22-16693R1Health professionals’ readiness to implement electronic medical recording system and associated factors in public general hospitals of Sidama region, EthiopiaPLOS ONE

Dear Dr. Kibruyisfaw,

Thank you for submitting your manuscript to PLOS ONE. After careful consideration, we feel that it has merit but does not fully meet PLOS ONE’s publication criteria as it currently stands. Therefore, we invite you to submit a revised version of the manuscript that addresses the points raised during the review process. Please submit your revised manuscript by October 30, 2022. If you will need more time than this to complete your revisions, please reply to this message or contact the journal office at plosone@plos.org. Please include the following items when submitting your revised manuscript:A rebuttal letter that responds to each point raised by the academic editor and reviewer(s). You should upload this letter as a separate file labeled 'Response to Reviewers'.A marked-up copy of your manuscript that highlights changes made to the original version. You should upload this as a separate file labeled 'Revised Manuscript with Track Changes'.An unmarked version of your revised paper without tracked changes. You should upload this as a separate file labeled 'Manuscript'.If applicable, we recommend that you deposit your laboratory protocols in protocols.io to enhance the reproducibility of your results. Protocols.io assigns your protocol its own identifier (DOI) so that it can be cited independently in the future. For instructions see: https://journals.plos.org/plosone/s/submission-guidelines#loc-laboratory-protocols. Additionally, PLOS ONE offers an option for publishing peer-reviewed Lab Protocol articles, which describe protocols hosted on protocols.io. Read more information on sharing protocols at https://plos.org/protocols?utm_medium=editorial-email&utm_source=authorletters&utm_campaign=protocols.

We look forward to receiving your revised manuscript.

Kind regards,

Humayun Kabir, MSc in Epidemiology

Academic Editor

PLOS ONE

Journal Requirements:

Reviewers' comments:

Reviewer's Responses to Questions

**Comments to the Author**

1. If the authors have adequately addressed your comments raised in a previous round of review and you feel that this manuscript is now acceptable for publication, you may indicate that here to bypass the “Comments to the Author” section, enter your conflict of interest statement in the “Confidential to Editor” section, and submit your "Accept" recommendation.

Reviewer #1: All comments have been addressed

Reviewer #2: (No Response)

2. Is the manuscript technically sound, and do the data support the conclusions?

Reviewer #1: Yes

Reviewer #2: Yes

3. Has the statistical analysis been performed appropriately and rigorously? 

Reviewer #1: Yes

Reviewer #2: Yes

4. Have the authors made all data underlying the findings in their manuscript fully available?

Reviewer #1: No

Reviewer #2: Yes

5. Is the manuscript presented in an intelligible fashion and written in standard English?

Reviewer #1: Yes

Reviewer #2: Yes

6. Review Comments to the Author

Reviewer #1: All the comments have been addressed. Therefore, I suggest to accept the manuscript. Furthermore, the underlying dataset could be checked, and probably a single excel sheet would be more aesthetic for the interested readers/researchers. If possible, provide the coded transcripts as the minimum underlying dataset for the qualitative part. The statistical issues could be re-checked by the expert before publication.

Reviewer #2: I would like to make a suggestion to the authors that in Table 3, they should include a footnote stating for which variables they accounted for adjusting their analysis.

7. PLOS authors have the option to publish the peer review history of their article (what does this mean?). If published, this will include your full peer review and any attached files.

Reviewer #1: No

Reviewer #2: **Yes: **Saifur Rahman Chowdhury

---

## [Author Response · Author response to Decision Letter 1]

30 Sep 2022

We would like to take a moment to thank the Editor and the peer reviewers for the constructive evaluation of our paper. We have corrected the manuscript as per the comments provided and we hope the manuscript meets the requirement for publication.

Sincerely,

Kibruyisfaw Weldeab Abore

Response to academic editor

Please review your reference list to ensure that it is complete and correct. If you have cited papers that have been retracted, please include the rationale for doing so in the manuscript text, or remove these references and replace them with relevant current references. Any changes to the reference list should be mentioned in the rebuttal letter that accompanies your revised manuscript. If you need to cite a retracted article, indicate the article’s retracted status in the References list and also include a citation and full reference for the retraction notice

Dear editor,

Thank you for taking the time to review our manuscript and for the constructive comment. After reviewing the reference list, although we have the booklet at hand, we noticed that one of the references booklets from ministry of health Ethiopia had no database to which readers could access the document. Therefore, we have removed this reference and updated it with a roadmap booklet that can be accessed from ministry of health E-library. 

Reviewer 1: All the comments have been addressed. Therefore, I suggest to accept the manuscript. Furthermore, the underlying dataset could be checked, and probably a single excel sheet would be more aesthetic for the interested readers/researchers. If possible, provide the coded transcripts as the minimum underlying dataset for the qualitative part. The statistical issues could be re-checked by the expert before publication.

I would like to extend my sincere gratitude on behalf of all authors for the constructive review we received and for the support towards the publication.

Reviewer 2: I would like to make a suggestion to the authors that in Table 3, they should include a footnote stating for which variables they accounted for adjusting their analysis.

I would like to extend my sincere gratitude on behalf of all authors for the constructive review we received and for the support towards the publication. We have amended the manuscript to include a foot note on Table 3 for the variables accounted for in the multivariable logistic regression.

---

## [Editor Report · Decision Letter 2]

6 Oct 2022

Health professionals’ readiness to implement electronic medical recording system and associated factors in public general hospitals of Sidama region, Ethiopia

PONE-D-22-16693R2

Dear Dr. Kibruyisfaw,

We’re pleased to inform you that your manuscript has been judged scientifically suitable for publication and will be formally accepted for publication once it meets all outstanding technical requirements.

Kind regards,

Humayun Kabir, MSc in Epidemiology

Academic Editor

PLOS ONE
---

## [Editor Report · Acceptance letter]

10 Oct 2022

PONE-D-22-16693R2 

Health professionals’ readiness to implement electronic medical recording system and associated factors in public general hospitals of Sidama region, Ethiopia 

Dear Dr. abore:

I'm pleased to inform you that your manuscript has been deemed suitable for publication in PLOS ONE. Congratulations! Your manuscript is now with our production department. 

Kind regards, 

on behalf of

Mr. Humayun Kabir 

Academic Editor

PLOS ONE